# Feasibility study of eye movement desensitisation and reprocessing (EMDR) in people with an at-risk mental state (ARMS) for psychosis: study protocol

Daniela Strelchuk  ,[1,2] Nicola Wiles,[1,2] Katrina M Turner  ,[1,2] Catherine Derrick,[1] Stan Zammit[1,3]

[1]Centre for Academic Mental Health, Population Health Sciences, Bristol Medical School, University of Bristol, Bristol, UK
[2]NIHR Biomedical Research Centre at University Hospitals Bristol and Weston NHS Foundation Trust and the University of Bristol, Bristol, UK
[3]Division of Psychological Medicine and Clinical Neuroscience, MRC Centre for Neuropsychiatric Genetics and Genomics, Cardiff University, Cardiff, UK

**Correspondence to**
Ms Daniela Strelchuk;
daniela.strelchuk@bristol.ac.uk

## ABSTRACT

**Introduction** Trauma can play an important role in the development of psychosis, yet no studies have investigated whether a trauma-focused psychological therapy could prevent the onset of psychosis in people at high risk of developing this condition. This study aims to establish whether it would be feasible to conduct a multicentre randomised controlled trial (RCT) to investigate the clinical and cost-effectiveness of eye movement desensitisation and reprocessing (EMDR) therapy to prevent the onset of psychosis in people with an at-risk mental state (ARMS).

**Methods/analysis** This is a single-arm trial with a nested qualitative study where all participants (target n=20) will be offered EMDR. Eligible participants are those who meet criteria for ARMS; have experienced a traumatic event before the onset of ARMS symptomatology; and have at least one symptom of post-traumatic stress disorder (PTSD). Participants will be followed up at 4, 8 and 12 months after the baseline assessment. The primary outcome measure is transition to psychosis, and secondary outcome measures include severity of psychotic symptoms, PTSD, depression, anxiety, impaired functioning, health status and resource use. The analysis will aim to establish the rates of recruitment and retention for a large-scale RCT. Interviews with therapists and patients will explore their views of the study and their experiences of delivering or receiving EMDR.

**Ethics and dissemination** This protocol has been approved by the South West-Cornwall and Plymouth Research Ethics Committee (Reference 18/SW/0037). Findings will be disseminated through journal publications, conference presentations and meetings with service users, their families, mental health professionals and commissioners.

**Trial registration number** ISRCTN31976295.

## INTRODUCTION

Psychotic illnesses are one of the leading causes of disability worldwide.[1] The onset of psychosis is often preceded by a prodromal phase, known as an at-risk mental state (ARMS) for psychosis, which is characterised

### Strengths and limitations of this study

► This is the first study to investigate the feasibility of conducting a large multicentre randomised controlled trial (RCT) of eye movement desensitisation and reprocessing (EMDR) for the prevention of psychosis in people with at-risk mental state.
► The study includes qualitative work to explore therapists and patients' views of the treatment offered/received.
► Therapy sessions follow an EMDR protocol specifically designed for people with psychosis.
► The decision to change the study design to a single-arm trial may limit our ability to draw robust conclusions about recruiting to an RCT.

by a series of non-specific or attenuated psychotic symptoms.[2] Approximately 22% of ARMS individuals will transition to psychosis within 1 year and 36% of them within 3 years.[3]

Various interventions have been used for the prevention of psychosis, but there is still limited evidence on the effectiveness of these approaches. Clinical guidelines recommend cognitive–behavioural therapy (CBT) as a first-line treatment.[4] However, while a meta-analysis of randomised controlled trials (RCT) showed that CBT has a moderate effect on transition to psychosis at 12 months of follow-up (relative risk (RR) 0.64, 95% CI 0.44 to 0.93), the evidence was of low to moderate quality and the effect was not maintained over the longer term (18 months: RR 0.55, 95% CI 0.25 to 1.19).[5]

Exposure to trauma is two to three times more common in ARMS or psychosis than in controls,[6–8] and post-traumatic stress disorder (PTSD) is also common, although not usually detected.[9] Despite strong evidence showing that trauma is a key factor in the development

BMJ

of psychosis,[10–12] no studies have yet investigated whether a trauma-focused therapy could prevent the onset of psychosis in ARMS.

Eye movement desensitisation and reprocessing (EMDR) is a trauma-focused therapy that has comparable efficacy to trauma-focused CBT (TF-CBT) in treating PTSD.[13–16] However, the two forms of therapy are different. While TF-CBT seeks to directly identify and restructure the negative beliefs, EMDR does this indirectly, relying more on the integration of emotional, cognitive, sensory, physical and contextual information to allow adaptive processing of traumatic memories. Unlike TF-CBT, EMDR does not require patients to give a detailed description of the trauma, which may be distressing. Also, unlike TF-CBT, EMDR does not involve homework, which may make it more suitable for people with a disorganised lifestyle, such as those with ARMS.

To date, there has been only one RCT comparing EMDR with treatment as usual (TAU) in patients with comorbid psychosis and PTSD. This study showed that EMDR improved PTSD, and also reduced paranoid delusions, but did not reduce auditory hallucinations at 6 months of follow-up.[17] However, the EMDR protocol did not specifically target psychotic symptoms. Furthermore, most participants had chronic forms of psychosis and hence psychotic symptoms might have been less amenable to treatment than more recent onset symptoms present in ARMS subjects.

## Objectives

The overarching aim of this study is to establish whether it would be feasible to conduct a large multicentre RCT to evaluate the clinical and cost-effectiveness of EMDR to prevent the onset of psychosis in people with an ARMS. The study has the following objectives:

1. To estimate the rate of recruitment and retention to inform the large-scale RCT.
2. To refine the eligibility criteria, screening and recruitment procedures.
3. To optimise the EMDR protocols and learn about the factors which affect the implementation of EMDR.
4. To explore patients and therapists' views of EMDR as a treatment for ARMS.
5. To investigate patients and therapists' views of the study design and study materials.
6. To understand what TAU consists of for patients with ARMS.

## METHODS AND ANALYSIS
### Study design

The study consists of a single-arm feasibility trial with a nested qualitative study, and is conducted in collaboration with the Early Intervention (EI) teams from the Avon and Wiltshire Mental Health Partnership NHS Trust (AWP). All participants will be offered EMDR, and will be assessed at baseline, 4, 8 and 12 months of follow-up as described below.

The study design outlined is the final version of the trial protocol. The original protocol was changed as a result of low recruitment. Recruitment challenges are described later.

## Inclusion and exclusion criteria
### Inclusion criteria

► Aged 16 years or over who are at risk of psychosis (as defined in the Comprehensive Assessment of At-Risk Mental States (CAARMS)).[2]
► Presence of at least one positive symptom of psychosis (perceptual abnormality, unusual thought content, non-bizarre ideas or disorganised speech) scored ≥3 on CAARMS.
► History of traumatic experience as defined in International Classification of Diseases 10th Revision (ICD-10) F43.1, occurring prior to onset of first positive symptom.
► Presence of one or more symptoms of reliving, avoidance, hyperarousal or cognitive distortions in relation to the traumatic experience (assessed using the PTSD Checklist for DSM-5 (PCL-5)) during the last month.[18]

### Exclusion criteria

► History of treated or untreated psychotic illness or learning disability.
► Currently taking antipsychotics.
► Currently receiving psychological therapy.
► Completed a trauma-focused psychological therapy in the last 2 years.
► Insufficient fluency in English.
► Lacking mental capacity to provide valid informed consent.

## Recruitment procedure

Participants will be identified via two routes.

### EI services

Researchers will identify clinical teams who are willing to identify potential participants via the following routes:

A. During routine clinic appointments—clinicians will introduce the study to patients who have met ARMS criteria within the past 12 months. Interested patients will be asked to complete section 1 of an Expression of Interest (EOI) form, and to provide their contact details. Completed EOI forms will be returned to the research team by secure email. A member of the research team will telephone interested individuals and, if the individual is willing to take part, agree a time and place for the researcher to meet with them to complete the eligibility assessment. Following the telephone call, the researcher will post the individual a full participant information sheet and a letter confirming the appointment.

Individuals who do not want to hear more about the study will be asked to complete section 2 of the EOI form, which asks for information on age, gender, reasons for non-participation and willingness to take part

in a short interview (see 'qualitative data collection' for more details).

B. Postal or telephone invitations from clinicians—if clinicians do not introduce the study to patients during the appointment, they will post patients with ARMS a brief information sheet about the study, an EOI form and a stamped addressed envelope. Clinicians may also introduce the study to patients over the telephone. If the patients are interested, the clinician will ask the patients for permission to share their contact details with the research team. The clinician will then fill in the 'Phone permission to contact' form and will email it to the research team. Recruitment will then proceed as outlined in (A).

C. Postal invitations sent out by National Institute for Health Research (NIHR) Clinical Research Network Clinical Studies Officers (CSO)—CSOs will prescreen potential participants using the electronic patient record system (RiO), and liaise with the care team regarding potentially eligible patients who will then be sent a brief information sheet and details on how to contact the research team if they are interested in the study. Patients who do not reply will be followed up by telephone by the CSO within a week of receiving the documents. If the patient is interested in the study, the CSO will take permission over the phone to share their contact details with the research team. Once the research team have received contact details, recruitment will proceed as outlined in (A).

### Everyone included within AWP

'Everyone Included' is a standard approach to research in AWP, whereby service users are routinely informed about relevant research opportunities. The AWP Research and Development department will send a 'Research Opportunity Letter' on behalf of the research team to potential participants briefly explaining the study and what it involves. A researcher will telephone individuals who are interested in the study and answer any questions. If the individual is willing to take part, an eligibility assessment will be arranged.

### Consent and baseline eligibility assessment

Individuals interested in taking part will be invited to a face-to-face appointment with a researcher to establish eligibility, answer questions about participation and obtain written informed consent.

The process of consenting young people (16 and 17 years old) will follow the Department of Health Reference Guide to Consent[19] and the Good Medical Practice Guidelines,[20] which provide the guide that all health professionals need to take into account in obtaining consent. Although young people may be more vulnerable than adults, they are presumed to be able to give consent to their treatment. As in the case of adults, if young people decline to take part in the study their decision will be fully respected and accepted.

In order to establish eligibility, patients will be asked to complete the following scales:
► Life Events Checklist for DSM-5.[21]
► Childhood Trauma Questionnaire.[22]
► PCL-5.[18]

Those who are eligible will be asked to provide written informed consent for their participation in the trial and sociodemographic information. Additional baseline data will be collected using the following quantitative measures:
► Severity of psychotic symptoms: CAARMS,[2] the Negative Scale of the Positive and Negative Syndrome Scale (PANSS),[23] the Psychotic Symptom Rating Scales (PSYRATS)[24] and the Community Assessment of Psychic Experiences (CAPE-42).[25]
► Severity of depression and anxiety: Patient Health Questionnaire (PHQ-9) for depression[26] and Generalized Anxiety Disorder Questionnaire (GAD-7) for anxiety.[27]
► Impaired functioning: Work and Social Adjustment Scale (WSAS).[28]
► Health status: Five-Level version of the EQ-5D (EQ-5D-5L).[29]
► Drug use: the Drug Abuse Screening Test (DAST-10).[30]

Data will also be collected on medication use via self-report measures. The baseline eligibility assessment will last about 90 min.

### Intervention: EMDR

Participants will receive up to 12 sessions of manualised, weekly, face-to-face EMDR therapy.[31] Treatment has been manualised by EMDR consultants from Lancashire and South Cumbria Care NHS Foundation Trust (LSCFT) and individualised to target psychotic symptoms. The treatment protocol was developed and evaluated through pilot cases with first-episode psychosis clients,[32] and further refined as part of an ongoing feasibility RCT funded by the NIHR (PB-PG-0317-20037; ISRCTN43816889). Therapeutic sessions will be delivered by EMDR UK and Europe trained therapists who will receive training based on the LSCFT manual for the study. The training will be tailored to therapists' needs, and will last between 4 and 8 hours. All EMDR therapists will have at least 1 year of experience of providing EMDR, and monthly supervision with an EMDR consultant will be in place. Each session will last approximately 90 min. The first two to four sessions will focus on establishing the therapeutic alliance, preparation for EMDR and stabilisation techniques, assessment and identification of targets, cognitions, emotions and bodily sensations. The following 8–10 sessions will focus on desensitisation, installation of positive cognitions and body scan for past, present and future stressful situations.

Written permission will be sought to audio record the therapeutic sessions. Ten per cent of the sessions will be randomly sampled and evaluated by accredited EMDR therapists independent of the study, to check treatment fidelity, using the EMDR fidelity checklist.[33] Therapists will be encouraged to liaise with participants' care

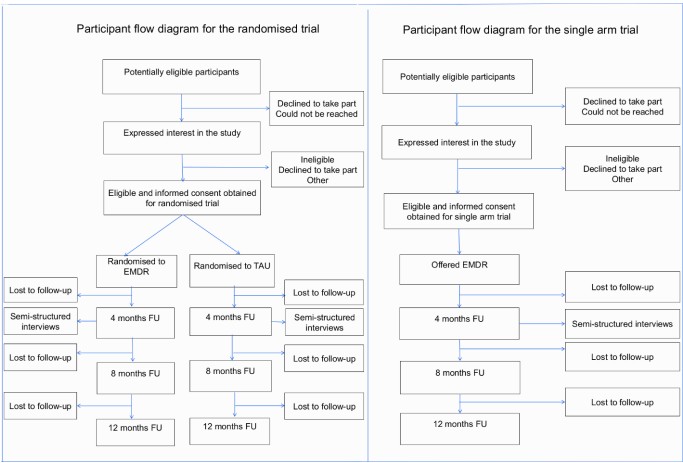

**Figure 1** Study flow chart for the feasibility trial (randomised and non-randomised components). EMDR, eye movement desensitisation and reprocessing; FU, follow-up; TAU, treatment as usual.

coordinators as this could improve participants' adherence to therapy. If a serious adverse event (SAE) occurs in relation to therapy, or participant has difficulties engaging with therapy to the point where therapy is no longer worthwhile, the therapy will be discontinued.

### Challenges to recruitment

The original feasibility study was designed as an RCT comparing EMDR with TAU. TAU in AWP is quite heterogeneous and consists of discharge back to the referrer, signposting to other services, CBT, family intervention or psychoeducation. Randomisation took place using a remote computerised randomisation service administered by the Bristol Trials Centre ensuring that allocations were concealed from the recruiting researcher. Randomisation was minimised by psychotic symptom severity (score on positive symptoms on CAARMS <11 or ≥11). However, following challenges to recruitment (described below), the study design was changed to a single-arm study.

During the first 7 months of the study (May to December 2018), we only recruited three participants (against a target of 21). This poor recruitment was primarily because the number of patients identified as ARMS by the EI teams within AWP was much lower than expected, and only three of the six EI teams we originally planned to recruit from were able to support the study due to financial constraints to managing ARMS clients. Therefore, from July 2019, all participants recruited to the study were offered EMDR. Figure 1 shows the flow of participants in the randomised and single-arm phases of the trial.

### Sample size and feasibility of recruitment

In the original protocol we had planned to recruit up to 40 participants. The initial sample size was informed by literature guidelines[34] and pragmatic considerations concerning potential difficulties accessing this population. The feasibility study would not be powered to detect important clinical differences between EMDR and TAU groups but would provide estimates of the completion and retention rates that would further assist in planning the recruitment for a future RCT.

Given the challenges with recruitment and modification of our study design (from a randomised to single-arm trial), we decreased our target sample size from 40 to 20 participants. This sample size will provide us with reasonable precision around estimates of percentages completing the intervention and follow-ups. For example, if 70% of those who started EMDR complete the intervention, the 95% CIs would be 45% and 88%. Similarly, if 75% of those who started EMDR are followed up to 12 months, the 95% CIs for the retention rate would be 50% and 91%.

### Outcome assessment

The follow-up schedule will involve face-to-face assessments at 4, 8 and 12 months after the baseline assessment. Each follow-up assessment will last about 60 min. The primary outcome will be collected at 12 months of follow-up, and the secondary outcome at 4, 8 and 12 months.

A 12-month follow-up will provide us with adequate data on retention rate that can be used to ensure that a definitive trial is adequately powered. If, in the definitive trial, the intervention was shown to be effective at 12 months, then further follow-up could examine whether effects are maintained over the longer term.

### Primary outcome

▶ Transition to psychosis: ICD-10 diagnosis of psychotic disorder from clinical records or via the CAARMS.

### Secondary outcome

▶ Severity of psychotic symptoms: CAARMS, PSYRATS, the Negative Scale of the PANSS and CAPE-42.
▶ Severity of PTSD symptoms: PCL-5.
▶ Severity of depression and anxiety: PHQ-9 and GAD-7.
▶ Impaired functioning: WSAS.
▶ Health status: EQ-5D-5L.
▶ Drug use: DAST-10.

- ► Medication use.
- ► Resource data use—we will ask questions about resource use including information on the use of primary and secondary care services, use of social services and disability payments received, time off school or work and productivity loss due to time off work or study.

If there are difficulties following up trial participants to obtain outcome data, researchers will prioritise collection of data on psychotic symptoms, PTSD symptoms and medication. Following the switch to single-arm design, it will no longer be possible to blind researchers to treatment allocation in terms of avoiding potential bias in outcome measurement.

### Quantitative data analysis

Quantitative data will be analysed in Stata V.15.[35] If there are no differences in baseline characteristics of those who were randomised and those who enter the single-arm study, we will combine data to generate summary statistics on: (1) the proportion consenting to take part in the study; (2) the proportion completing the baseline assessment and agreeing to take part in the study; (3) the number of EMDR sessions attended and the proportion completing eight or more sessions (regarded as an adequate dose of therapy); (4) the proportion completing the follow-up assessments. Feasibility outcomes such as recruitment and retention rates will be calculated with 95% CIs using the exact binomial method.

We will also report the proportion of individuals who transitioned to psychosis at 12 months, and the CIs for the effect size will be calculated to assist the planning of a future RCT. We will also compare the continuous outcomes such as the severity of psychotic symptoms, depression, anxiety, PTSD symptoms and quality of life to show the frequency of data completion, mean and SD for the three time points.

### Qualitative data collection

We will conduct in-depth individual interviews with all the therapists and participants who participate in the trial. In addition, up to 10 individuals who declined to take part in the trial will be interviewed to explore their views and understanding of the study and EMDR, and their reasons for declining to take part.

To encourage participation, and because well-planned interviews can gather the same material as those conducted face to face,[36] therapists and study participants will be given the choice of being interviewed in person or over the telephone. Therapists will be interviewed within a month of finishing intervention delivery. The interviews will explore therapist views and experiences of delivering EMDR to patients with ARMS, how treatment could be better tailored to this patient group and what resources they would need if the intervention was evaluated in a large-scale RCT. The interviews will also be used to explore their views of the training and supervision received.

Interviews with participants who completed treatment or stopped early will explore their views and experiences of the treatment received, identify how treatment delivery could be refined to increase its acceptability and how study materials could be improved. Participants who completed the study will be interviewed within a month of completing their 4-month follow-up, and those who stopped early will be interviewed within a month of doing so.

Topic guides will be developed for each set of interviews to ensure consistency across data collection. The therapist and study participant guides will be developed in parallel to ensure key areas are discussed with both practitioners and patients. This will help compare findings across the interviews, highlighting similarities and differences between the views and experiences of patients and practitioners, and increasing the confidence with which study conclusions can be drawn. With interviewee consent, the interviews will be audio recorded and transcribed verbatim.

### Qualitative data analysis

All interviews will be fully transcribed and anonymised. Data collection and analysis will proceed in parallel, so that insights from earlier data can shape later data collection, and enable the team to establish when data saturation has been reached. The data gathered will be analysed thematically. Transcripts will be independently coded by different researchers, who will then meet to discuss their coding and interpretation of the data. This will help control for researcher bias and may lead to the coding frame being revised or defined more clearly. Transcripts will then be imported into the software package NVivo V.12[37] to allow electronic coding and retrieval of data. Once all the transcripts have been coded, data will be analysed using an approach similar to framework analysis.[38]

### Additional qualitative work to address issues with recruitment and explore treatment options in ARMS populations

As the study progressed, to explore why recruitment had been so low, additional objectives were incorporated. These were:

1. To explore how potential patients with ARMS are identified and managed in primary and secondary care services.
2. To identify referral routes between primary and secondary care.
3. To explore other researchers' experiences of recruiting patients with ARMS to research studies in the UK. This additional qualitative work will entail:
A. Interviews with general practitioners (GPs), clinicians from Primary Care Liaison Services and EI services.
B. Interviews with patients who did not participate in the interventional part of the study but who have been identified as ARMS by the EI teams we are recruiting from.

C. Interviews with researchers who have been involved in recruiting patients with ARMS to research studies in the UK.

## Summary of changes to the original feasibility study

There have been a number of changes to the original feasibility study. These are summarised below:

A. Change of study design from randomised trial to single-arm trial.
B. Decrease in sample size from 40 to 20 participants.
C. Change in the measure used to assess the severity of psychotic symptoms at baseline and follow-up assessments from the Positive Scale of the PANSS to the CAARMS, as the latter is more sensitive.
D. Extending the time between patients being identified as ARMS to the time when they can be recruited into the study from 3 to 12 months.
E. Additional interviews with GPs, other clinicians from primary and secondary care services, patients with ARMS and researchers to better understand identification and management of patients with ARMS, including routes into primary and secondary services.

## Study management
### Data management

Completed questionnaires will be stored securely in compliance with University of Bristol Data Security policies and the General Data Protection Regulation. Personal details will be entered onto a secure database held on the University of Bristol server, and non-identifiable data will be entered onto a secure web-based database. Data collected on paper will be identifiable only by patient identification number. Information capable of identifying individuals will be held on the database with passwords restricted to authorised study staff only.

Data will be stored in the University of Bristol research data repository, and will be available on reasonable request.

### Patient and public involvement

Patient and public involvement (PPI) played an important role in this study. This research idea was initially presented at a PPI event which brought together people with lived experiences of psychosis, their carers, clinicians, academics and commissioners. Further on, this idea was presented at the Hearing Voices Network, and through these contacts we established a PPI group who gave us important feedback on the research question, the intervention and which measures to use. We anticipate that the group, which is still active, will: (1) ensure that the views of patients who have had a psychotic illness are heard at every stage of the study; (2) inform any developments of the intervention to make sure these are acceptable to the target population, (3) ensure patient documentation is easy to understand to maximise response and retention rates; and (4) help interpret and disseminate study findings.

## MONITORING AND ADVERSE EVENTS
### Adverse events

We use the recommendations for defining and reporting adverse events and harm as outlined by Parry et al.[39] All SAEs will be reported to the chief investigator and the Trial Steering Committee (TSC). All SAEs of a related and unexpected nature will be reported to the main Research Ethics Committee (REC), in accordance with any procedures of the sponsor. The decision to audit the trial rests with University Hospitals Bristol.

### Trial oversight

A TSC will have independent oversight of the study, meeting at six monthly intervals during the course of the trial. The TSC will also take on data monitoring role for this feasibility study.

## DISCUSSION

To our knowledge, this is the first study to investigate the feasibility of conducting a large RCT to establish whether EMDR could prevent the onset of psychosis in people with an ARMS. As psychotic illnesses usually occur in early adulthood and disrupt a key period of professional and social development, preventing psychosis would improve patients' outcomes, and also decrease the substantial economic burden of this condition.

The results of this feasibility trial will provide us with a better understanding of this patient group, and will provide us with information that will be used to inform whether progression to a full trial would be feasible. The nested qualitative study will address issues of patient and therapist views of the intervention, and provide new insights into treatment pathways and experiences of this patient group. If shown to be feasible, a full-scale RCT will be conducted to examine the clinical and cost-effectiveness of EMDR to prevent the onset of psychosis in ARMS.

## ETHICS AND DISSEMINATION

This protocol has been approved by South West-Cornwall and Plymouth REC (formerly South West-Exeter REC), Reference 18/SW/0037, 29.03.2018.

The results of this study will be disseminated through publications and conferences. Findings from this feasibility trial will be published in psychiatry journals compliant with NIHR Open Access policy, and presented to clinical staff, service users and relevant clinical commissioning groups.

### Feasibility trial status

Recruitment to this study opened in May 2018, and closed in May 2020. At the time of writing (July 2020), 14 participants have been recruited to the study. We expect to finish follow-up data collection by May 2021.

This article describes protocol version 4, dated 2 August 2019. When all regulatory approvals had been received,

the protocol was version 1, dated 11 January 2018. Since then, two major amendments have been made with changes to incorporate additional qualitative work, decrease the sample size and change the study design from an RCT to a single-arm trial. The latter change took place in July 2019 prior to which we had recruited six participants.

**Acknowledgements**  We would like to thank Professor Jonathan Bisson, Professor Peter Jones and Dr Fiona Warren for overseeing the trial as part of our TSC group. The authors would also like to thank the members of our PPI group (Mr Bradley Jones, Ms Claire Barnard and Joanna Gigon) for their feedback on the research question, intervention and measures to use, comments on the supporting documentation and suggestions on how to improve recruitment and increase retention to the study.

**Contributors**  SZ, NW, KMT and DS designed the study. SZ is the grant holder and chief investigator. SZ, NW and KMT oversaw the scientific rigour of the study. NW provided expertise in the management and analysis of the study. KMT oversaw the nested qualitative study. DS coordinated, implemented and ran the trial. CD assisted with running the trial. All authors contributed to refinement of the study protocol and approved the final manuscript.

**Funding**  This study was funded by the NIHR Biomedical Research Centre at University Hospitals Bristol NHS Foundation Trust (BRC-1215-20011) and the University of Bristol. The sponsor of the study is the University of Bristol.

**Disclaimer**  The views expressed in this publication are those of the author(s) and not necessarily those of the NHS, the National Institute for Health Research or the Department of Health. The sponsor and study funders had no authority over the running of the trial, and all decisions rest with the trial team.

**Competing interests**  None declared.

**Patient and public involvement**  Patients and/or the public were involved in the design, or conduct, or reporting, or dissemination plans of this research. Refer to the Methods section for further details.

**Patient consent for publication**  Not required.

**Provenance and peer review**  Not commissioned; externally peer reviewed.

**ORCID iDs**
Daniela Strelchuk http://orcid.org/0000-0002-1634-2801
Katrina M Turner http://orcid.org/0000-0002-6375-2918

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
