## [Reviewer comments · BMJ Open]

ARTICLE DETAILS

TITLE (PROVISIONAL)	A feasibility study of eye-movement desensitization and reprocessing (EMDR) in people with an at-risk mental state (ARMS) for psychosis: study protocol
AUTHORS	Strelchuk, Daniela; Wiles, Nicola; Turner, Katrina; Derrick, Catherine; Zammit, Stan

VERSION 1 – REVIEW

REVIEWER	Cinzia Perlini Department of Neurosciences, Biomedicine and Movement Sciences, Section of Clinical Psychology, University of Verona, Verona, Italy
REVIEW RETURNED	23-Apr-2020

GENERAL COMMENTS	This is a study research protocol aiming at inform the feasibility of a future multicentre RCT. Focus of the present protocol is the implementation of an ad-hoc Eye Movement Desensitisation and Reprocessing (EMDR) program to prevent the transition to psychosis in a group of 20 at risk mental state (ARMS) subjects. Moreover, a qualitative investigation on patients' and therapists' views of EMDR will be included. The protocol is detailed and appropriately described. Despite that, I have some major and minor issues that should be addressed before the publication of the manuscript. Major issues: - In available literature and in the Introduction Authors clearly stated that the rate of the transition to psychosis in ARMS population is 22% within one year and 36% within 3 years. According to Fusar-Poli et al., 2012 Archives of General Psychiatry, 69 (3) (2012), pp. 220-229, rates of transition to a psychotic episode from the ARMS high-risk state were 18% after 6 months of follow-up, 22% after 1 year, 29% after 2 years, and 36% after 3 years. Since the transition to psychosis represents the primary outcome of the study and the measure of the effectiveness of EMDR, it is not clear the choice of 12 months as the longest period of follow up. How Authors can exclude a late transition to psychosis? And also, Authors wrote that the effect of CBT on the transition to psychosis is not maintained over 18 months, thus suggesting the general need to extend follow ups assessments.- In the manuscript it is specified that ARMS people aged 16 years or over will be recruited but no details on the informed consent for subjects under the age of 18 are given. Minor issues: - Authors should write the full name of scales and questionnaire+acronyms the first time they cite them in the manuscript (then acronyms only)
--

	 - In the description of the EMDR intervention, at least one reference to a paper/chapter/book describing the phases of the EMDR therapy should be provided - The version of the Stata software should be specified and a reference provided. The same for the NVivo software.
--	--

VERSION 1 – AUTHOR RESPONSE

Reviewer: 1

This is a study research protocol aiming at inform the feasibility of a future multicentre RCT. Focus of the present protocol is the implementation of an ad-hoc Eye Movement Desensitisation and Reprocessing (EMDR) program to prevent the transition to psychosis in a group of 20 at risk mental state (ARMS) subjects. Moreover, a qualitative investigation on patients' and therapists' views of EMDR will be included.

The protocol is detailed and appropriately described. Despite that, I have some major and minor issues that should be addressed before the publication of the manuscript.

Major issues:

- In available literature and in the Introduction Authors clearly stated that the rate of the transition to psychosis in ARMS population is 22% within one year and 36% within 3 years. According to Fusar-Poli et al., 2012 Archives of General Psychiatry, 69 (3) (2012), pp. 220-229, rates of transition to a psychotic episode from the ARMS high-risk state were 18% after 6 months of follow-up, 22% after 1 year, 29% after 2 years, and 36% after 3 years. Since the transition to psychosis represents the primary outcome of the study and the measure of the effectiveness of EMDR, it is not clear the choice of 12 months as the longest period of follow up. How Authors can exclude a late transition to psychosis? And also, Authors wrote that the effect of CBT on the transition to psychosis is not maintained over 18 months, thus suggesting the general need to extend follow ups assessments.

We agree that long-term outcomes are important. However, it is challenging to maintain a high level of follow-up amongst trial participants over the longer term, and we think that, for a definitive trial, a 12-month follow-up is an acceptable time period that would allow us to observe a reasonable number of transitions to psychosis, and as such, a large-scale trial would be powered accordingly. In order to assist with the planning of such a trial, we designed this feasibility study with a 12-month follow-up as this would provide data on the likely retention rate. If, based on the results of this feasibility study, it was feasible to undertake a large-scale RCT, and if the intervention were found to be effective over a 12-month period, then we would look to seek additional funding to evaluate the longer-term effectiveness of the intervention.

- In the manuscript it is specified that ARMS people aged 16 years or over will be recruited but no details on the informed consent for subjects under the age of 18 are given.

Thank you for the very helpful comment. We have now added the following text under the “Consent and baseline eligibility assessment” (page 6, paragraph 5).

The process of consenting young people (16 and 17 years old) will follow the Department of Health Reference Guide to Consent[1] and the Good Medical Practice Guidelines,[2] which provide the guide that all health professionals need to take into account in obtaining consent. Although young people

may be more vulnerable than adults, they are presumed to be able to give consent to their treatment. As in the case of adults, if young people decline to take part in the study their decision will be fully respected and accepted.

Minor issues:

- Authors should write the full name of scales and questionnaire+acronyms the first time they cite them in the manuscript (then acronyms only)

We have now written the full name and acronyms of all scales and questionnaires:

- *Life-events Checklist for DSM-5 (LEC-5)[3]*
- *Childhood Trauma Questionnaire (CTQ)[4]*
- *Severity of psychotic symptoms: ... the Negative Scale of the Positive and Negative Syndrome Scale (PANSS),[5] the Psychotic Symptom Rating Scales (PSYRAT)[6] and the Community Assessment of Psychic Experiences (CAPE-42).[7]*
- *Health status: Five-level version of the EQ-5D (EQ-5D-5L)*

- In the description of the EMDR intervention, at least one reference to a paper/chapter/book describing the phases of the EMDR therapy should be provided

We have now added a reference for the EMDR therapy (page 7, paragraph 4).

Participants will receive up to 12 sessions of manualized, weekly, face-to-face EMDR therapy.[8]

- The version of the Stata software should be specified and a reference provided. The same for the NVivo software.

We have now specified the version of STATA and NVivo software, and also provided references.

Quantitative data will be analysed in Stata version 15.[10] (page 10, paragraph 1)

Transcripts will then be imported into the software package NVivo version 12 [11] (page 11, paragraph 2)

- 1 Department of Health and Social Care. Reference guide to consent for examination or treatment (second edition). 2009.<https://www.gov.uk/government/publications/reference-guide-to-consent-for-examination-or-treatment-second-edition> (accessed 3 May 2020).
- 2 General Medical Council. https://www.gmc-uk.org/static/documents/content/Good_medical_practice_-_English_1215.pdf. 2013.
- 3 Gray MJ, Litz BT, Hsu JL, *et al.* Psychometric properties of the life events checklist. *Assessment* 2004;**11**:330–41. doi:10.1177/1073191104269954
- 4 Bernstein DP, Fink L. *Childhood trauma questionnaire : a retrospective self-report : manual*. Orlando: : Psychological Corporation 1998.
- 5 Kay SR, Fiszbein A, Opler LA. The positive and negative syndrome scale (PANSS) for schizophrenia. *Schizophr Bull* 1987;**13**:261–76. doi:10.1093/schbul/13.2.261
- 6 Haddock G, McCarron J, Tarrier N, *et al.* Scales to measure dimensions of hallucinations and delusions: The psychotic symptom rating scales (PSYRATS). *Psychol Med* 1999;**29**:879–89. doi:10.1017/s0033291799008661

- 7 Stefanis NC, Hanssen M, Smirnis NK, *et al.* Evidence that three dimensions of psychosis have a distribution in the general population. *Psychol Med* 2002;**32**:347–58.
doi:10.1017/s0033291701005141
- 8 Shapiro F. *Eye Movement Desensitization and Reprocessing (EMDR) Therapy: Basic Principles, Protocols and Procedures*. 3rd ed. New York, NY: : The Guilford Press 2018.
- 9 Parnell L. *A therapist's guide to EMDR. Tools and techniques for successful treatment*. W.W.Norton & Company 2007.
- 10 StataCorp. Stata Statistical Software: Release 15. 2017.
- 11 QSR International Pty Ltd. NVivo (Version 12). 2018.<https://www.qsrinternational.com/nvivo-qualitative-data-analysis-software/home>